# Ultra-Flexible Organic Photovoltaics with Nanograting Patterns Based on CYTOP/Ag Nanowires Substrate

**DOI:** 10.3390/nano10112185

**Published:** 2020-11-02

**Authors:** Soo Won Heo

**Affiliations:** Nanomaterials and Nanotechnology Center, Electronic Convergence Division, Korea Institute of Ceramic Engineering and Technology (KICET), 101, Soho-ro, Jinju-si, Gyeongsangnam-do 52851, Korea; soowon.heo@kicet.re.kr; Tel.: +82-55-792-2679

**Keywords:** ultra-flexible OPVs, Ag nanowires, solution-processed ultrathin substrate, mechanical property

## Abstract

In this study, we developed a method for fabricating ultrathin polymer substrates that can be used in ultra-flexible organic photovoltaics (OPVs) via a non-vacuum process using cyclic transparent optical polymer. In addition, a Ag nanowire network layer was used as a transparent electrode in a solution process. All processes were conducted on large area via spin coating. The power conversion efficiency (PCE) of the ultra-flexible OPV improved by 6.4% compared to the PCE of the ITO/Glass-based OPV. In addition, the PCE of the OPV increased to 10.12% after introducing nanostructures in the ZnO and photoactive layers. We performed 1000 cycles of compression/relaxation tests to evaluate the mechanical properties of the ultra-flexible OPV, after which, the PCE remained at 67% of the initial value. Therefore, the developed OPV system is suitable as a power source for portable devices.

## 1. Introduction

Organic photovoltaics (OPVs) with a thickness of less than 5 µm are being actively studied as next-generation power sources for mobile applications and biosensors because they are very thin, light, and exhibit very high power-to-weight ratios [1,2,3,4,5,6]. An important factor in fabricating ultra-flexible OPVs is the development of ultra-flexible polymer substrates and the application of suitable transparent electrode materials.

Many researchers have applied parylene, polydimethylsiloxane (PDMS), and transparent polyimide (PI) as ultrathin polymer substrate materials in ultra-flexible OPVs [1,2,3,4,5,6]. In the case of parylene, chemical vapor deposition is essential, which makes it difficult to manufacture a large-area device and establish a continuous process [7,8,9]. For PDMS, solution processes such as spin coating are possible, but the solvent resistance such as acid etchant and alkaline developer is weak, making it difficult to withstand the etching process of transparent conductive oxides such as indium tin oxide (ITO) [10,11]. Furthermore, the transmittance of transparent PI is lower than those of parylene and PDMS, and its chemical resistance and coefficient of thermal expansion values are low [12,13,14]. Therefore, a more in-depth research is required to develop an ultra-flexible transparent substrate for OPV that satisfies the conditions as mentioned above.

ITO is one of the most widely used transparent electrodes for ultra-flexible OPVs owing to its low sheet resistance and transmittance of more than 90% [15]. However, ITO exhibits very rigid properties owing to its high crystallinity. This characteristic deteriorates the mechanical properties of ultra-flexible OPVs. Therefore, it is very important to introduce material for a transparent electrode with excellent mechanical properties, high transparency, and ease of fabrication.

To overcome this problem, we introduced an amorphous perfluoro-polymer (cyclic transparent optical polymer, CYTOP, AGC chemicals) as a substrate material and fabricated an ultrathin polymer substrate via a solution process [16]. CYTOP has an extremely high transparency, of which the visible light transmission ratio is more than 95% or more, with an amorphous structure completely different from existing fluoropolymers. In addition, it has excellent chemical resistance and moisture-proof property, so it can be applied to ultrathin substrates for ultra-flexible OPVs.

A self-assembly monolayer (SAM) was formed using trichloro(perfluorooctyl)silane (FOTS) to easily delaminate ultra-flexible OPVs on a supporting substrate (bare glass). CYTOP introduced an ultrathin polymer substrate that showed strong solvent resistance to various organic and polar solvents. Thus, we could apply the etching process of the transparent electrode to the ultrathin substrate.

In addition, Ag nanowires (NWs) were introduced as transparent electrodes to improve the mechanical properties of the ultra-flexible OPVs [17,18]. As the number of spin-coatings of the Ag NWs solution increased, the transmittance and sheet resistance of the transparent electrode reached 88% and 40 ohm/sq, respectively. Moreover, 1D nanograting patterns were introduced on both the hole transport layer such as poly(3,4-ethylenedioxythiophene) polystyrene sulfonate (PEDOT: PSS) and the photoactive layer for the first time to increase the power conversion efficiency (PCE) of the ultra-flexible OPV. The PCE improved from 8.93%(rigid structure without pattern) to 10.12% due to an increase in the short circuit current density (*J*_SC_) by introducing the 1D nanograting patterns.

## 2. Materials and Methods

### 2.1. SAM Treatment Process

To decrease the surface energy of the supporting substrate, we introduced a perfluoro silane-based SAM, viz. FOTS (98%, Merck), which can easily delaminate ultra-flexible OPVs from the supporting substrate. A UV/ozone-treated template (30 min, Ahtech LTS) and a small chalet were placed in a 300-mL container. Then, 30 µl of FOTS was dropped in the chalet, the lid was closed, and the container was placed in a vacuum chamber at 5 × 10^−3^ Torr for 1 h.

### 2.2. Fabrication of Ultrathin Substrate

A 1-µm-thick CYTOP (CTX-809AP2, AGC chemicals) layer was spin-coated onto the SAM-treated supporting substrate at 2000 rpm for 60 s, pre-annealed for 10 min at 80 °C to prevent bubbling of the surface of CYTOP, and annealed for 30 min at 150 °C on the hotplate. Moreover, a solvent-resistance test was conducted using various organic and polar solvents.

### 2.3. Fabrication of Ag NWs Transparent Electrode

Ag NWs (10 mg/mL, Novarials NovaWire-Ag-A70-IPA) were spin-coated onto the CYTOP ultrathin polymer layer. The number of spin-coatings of the Ag NWs solution increased from 1 to 3. To form a pattern of Ag NWs based transparent electrode, photoresist (AZ 1512, Microchem) was spin-coated in two steps (500 rpm for 20 s, 3000 rpm for 60 s), and soft bake was performed at 90 °C for 1 min on a hot plate. And then, prepared sample was exposed to UV light for 8 s using mask aligner (UFM-50110R, Yamashita Denso). The exposed photoresist was removed by an alkaline tetra methyl ammonium hydroxide developer such as AZ300 (Microchem). The sample was baked at 110 °C for 1 min after development to further cross-link the phenolic resin in the unexposed resist. The developed sample was etched by etchant for Ag NWs (Pure Etch GNW300, Hayashi Pure Chemical).

The transmittance and sheet resistance of the Ag NWs transparent electrodes were measured using a UV-Vis spectrophotometer (V-670, JASCO) and contact-type 4-point probe (NDK), respectively.

### 2.4. Fabrication of Ultra-Flexible OPVs with and without 1D Nanograting Patterns

A 100-nm-thick layer of the Ag NWs transparent electrode was patterned via photolithography and a wet etching process. Cr/Au electrodes (2.5-nm-thick Cr and 100-nm-thick Au) were prepared as contact pads onto the Ag-NWs layer using a thermal evaporator at a pressure below 5 × 10^−5^ Torr. Subsequently, a 40-nm-thick PEDOT: PSS layer was introduced as the hole transporting layer.

To introduce a 1D nanograting pattern to both the PEDOT: PSS and photoactive layers, we prepared a 1D nanograting-patterned PDMS mold [19]. The polycarbonate cover layer of a blank DVD-RW (Sony) was manually peeled off with a surgical knife to reveal the grating pattern. Then, we washed a DVD-RW polycarbonate master mold with isopropyl alcohol to remove the dye layer from the nanograting pattern and baked it on a hotplate for 10 min at 70 °C. The UV-curable PDMS mixture (X-34-4184-A/B, mixing ratio 1:1, Shin-Etsu) was degassed in a vacuum chamber for 10 min. The degassed PDMS mixture was poured onto the master mold, and the sample was degassed again for 10 min. The sample was irradiated with UV light (365 nm) for 3 h, and the 1D nanograting-patterned PDMS mold was peeled off the surface of the master mold. We achieved a 1D nanograting patterned PDMS mold with a period of 760 nm and a depth of 100 nm.

After spin-coating the PEDOT: PSS solution, we gently placed the patterned PDMS mold on top of the PEDOT: PSS layer within 5 s and delaminated the PDMS mold after 60 s. This procedure was optimized based on a systematic study of the stamping timing and thickness of the PEDOT: PSS layer. The average groove depth and period of the patterned PEDOT: PSS layer were 35 nm and 760 nm, respectively.

To prepare the photoactive layer, we dissolved a mixture of poly [4,8-bis(5-(2-ethylhexyl)thiophen-2-yl)benzo[1,2-b;4,5-25b’]dithiophene-2,6-diyl-alt-(4-octyl-3-fluorothieno [3,4-b]thiophene)-2-carboxylate-2-6-diyl]26 (PBDTTT-OFT) as an electron donor polymer and [6,6]-phenyl-C71-butyric acid methyl ester (PC_71_BM) as an electron acceptor small molecule (1:1.5, 10 mg:15 mg, Figure 5d,e) in 1 mL chlorobenzene with 3 vol% 1,8-diiodooctane (98%, Merck). The solution was spin-coated on the samples with and without a patterned PEDOT: PSS layer at 800 rpm for 7 s in a dry N_2_-filled glove box to form a 110-nm-thick layer.

To form a 1D nanograting pattern on the photoactive layer, we placed the patterned PDMS mold on top of the photoactive layer without additional pressure (period: 760 nm, depth: 55 nm). There were no visible remaining photoactive materials on the surface of the PDMS mold after peeling off.

The thin films were transferred into a thermal evaporator to introduce the metal electrode. A Ca electron-transporting layer (0.1 Å s^−1^, 20 nm) and Al electrodes (0.5 Å s^−1^, 100 nm) were deposited via thermal evaporation under a high vacuum (below 5 × 10^−5^ Torr) using a metal mask.

### 2.5. Measurements

The film thicknesses of the PEDOT: PSS and photoactive layers were determined using a surface profiler (DEKTAK 6M, Bruker). The water contact angle was measured using the DAS 100 (KRUSS) measurement system. Absorption and transmission were measured using a UV-Vis spectrophotometer (V-670, JASCO). The surface morphology images of the 1D nanograting patterned PEDOT: PSS and photoactive layers were observed via atomic force microscopy (5400, Agilent Technologies, Santa Clara, CA, USA) in tapping mode.

The J-V characteristics of the devices were measured using a source measurement unit (2400, Keithley, Cleveland, OH, USA) under simulated solar illumination (AM 1.5 G, 100 mW cm^−2^) from a solar simulator based on a 150-W Xe lamp (PEC-L11, Peccell Technologies, Yokohama, Japan). The light intensity was calibrated using a standard silicon solar cell (BS520, Bunkoh-Keiki, Tokyo, Japan). The active area of each device was defined with a 0.12 cm^2^ metal photomask. The external quantum efficiency (EQE) of each device was measured with monochromatic light (SM-250F, Bunkoh-Keiki, Tokyo, Japan).

## 3. Results and Discussion

Figure 1 shows a schematic illustration of the ultrathin polymer substrate with a Ag NW-based transparent electrode via the solution process. The detailed procedure is shown in the experimental section (see Section 2.1, Section 2.2 and Section 2.3).

SAM treatment was performed using FOTS to easily delaminate the ultra-flexible OPVs from the supporting substrate. As shown in Figure 2, the water contact angle of the FOTS-treated supporting substrate increased to 106.2°. The substrate changed to hydrophobic, which suggested that the surface energy of the SAM-treated supporting substrate was lower than that of the untreated one.

Figure 3a shows the transmittance of the FOTS-treated supporting substrate, which was similar to that of the bare supporting substrate. This suggested that FOTS was successfully formed on the supporting substrate as a monolayer. Moreover, the transmittance of the CYTOP-coated substrate was slightly higher than that of the bare substrate. The refractive index of the CYTOP was 1.34, which was introduced on the supporting substrate (refractive index: 1.5), thereby affecting the improvement of the transmittance. In addition, a solvent-resistance test was conducted on the CYTOP ultrathin substrate (Figure 3b). As shown in Figure 3c, the CYTOP ultrathin substrate showed strong solvent resistance to various organic solvents, polar solvents such as IPA, and deionized water. Therefore, as shown in Figure 4, we were able to pattern the Ag NW-based transparent electrode introduced on the CYTOP and secure the lithography process with a resolution of 5 µm using AZ1512 positive photoresist (Microchem). Figure 3d shows the transmittance and sheet resistance of the Ag NW-based transparent electrode with respect to the number of spin-coatings on the CYTOP ultrathin substrate. The Ag NWs transparent electrode that was coated once, including the supporting glass, showed 90% transmittance in the visible wavelength region (380–780 nm). As the number of coatings of the Ag NWs increased, both the transmittance and sheet resistance decreased. The sheet resistance of the Ag NWs that was coated three times was 40 ohm/sq. Moreover, the transmittance of the Ag NW-based transparent electrode reached 88%, it was suitable value for the application in OPVs. To confirm the uniformity of the Ag NWs transparent electrode coated on the CYTOP ultrathin substrate, the transmittance of five regions was measured and compared. As shown in Figure 3e, the transmittances of all five positions converged into one form. This suggested that high-quality Ag NWs transparent electrodes can be applied to the CYTOP ultrathin substrate. Figure 3f presents the optical properties of various structures of transparent electrodes. ITO/glass showed a transmittance of 88% in the visible spectrum, and the transmittance of Ag NWs/CYTOP/glass decreased by 14% to 77%. Conversely, the transmittance of the Ag NWs/CYTOP ultrathin substrate delaminated from the supporting substrate increased to 86%. The transmittance was improved because of the low refractive index of CYTOP and the fact that the interference between CYTOP and the supporting substrate was reduced after the removal of the supporting substrate.

We fabricated OPVs incorporating nanostructures to confirm that substrates fabricated with a non-vacuum process can increase the photocurrent cumulatively without optical interference. In this case, since the Ag NWs was used as the transparent electrode, PEDOT: PSS was introduced as the hole transport layer, unlike the device structure applied in our previous study [19]. Figure 5 exhibits a schematic illustration of the fabrication of ultra-flexible OPVs with a 1D nanograting pattern. We introduced 1D nanograting patterns to both the PEDOT: PSS and photoactive layers in OPVs via soft-nanoimprinting lithography using a PDMS mold. The ultra-flexible device structure was CYTOP (3 µm)/Ag NWs (150 nm)/PEDOT: PSS (with or without the pattern, ~40 nm)/photoactive (with or without the pattern, 110 nm)/Ca (20 nm)/Al electrode (100 nm). The pattern pitch of the 1D nanograting pattern applied to the OPVs was 760 nm. After the vacuum deposition of Ca and Al, the grating pattern remained visible on the surface of the Al electrode because the Ca/Al layer was thin enough to replicate the surface morphology.

Figure 6 presents the current density-voltage (*J*-*V*) characteristics of PBDTTT-OFT: PC_71_BM-based OPVs fabricated with the three different structures under 100 mW cm^−2^ air mass 1.5 global (AM 1.5G) illumination. The *J*_SC_, open-circuit voltage (*V*_OC_), fill factor (*FF*), and PCE of the devices are summarized in Table 1. After the introduction of the ultrathin polymer substrate, *J*_SC_ increased from 17.18 mA cm^−2^ in the reference device (ITO/Glass) to 17.76 mA cm^−2^ (Figure 6a). Consequently, the PCE increased from 8.93% to 9.50%. Moreover, the introduction of the nanograting patterns onto both the ZnO and photoactive sides further increased the *J*_SC_ to 18.40 mA cm^−2^ while maintaining similar *V*_OC_ values. This cumulative enhancement in *J*_SC_ resulted in a PCE of 10.12%. To clarify this, we calculated the *s*eries resistance (*R*_S_) and shunt resistance (*R*_Sh_) from the J-V curves (Table 1). The introduction of nanograting patterns slightly decreased the *R*_S_ values compared with the reference device, whereas it increased the *R*_Sh_ values [19].

To confirm the origin of the increase in *J*_SC_, we measured the EQE of the OPVs (Figure 6b). The light interference was reduced because of the optical properties of the Ag NWs/CYTOP ultra-flexible substrate (Figure 3f), which improved the EQE. Specifically, the EQE in the range of 300–420 nm increased. In addition, anti-reflection and light-guide effects appeared owing to the nanograting pattern that was introduced to the PEDOT: PSS layer. Therefore, the EQE increase in the 300–400 nm region was remarkable. Moreover, the optical path length improved because of the nanostructure introduced in the photoactive layer, and the photon absorption property in the range of 450–750 nm was drastically increased. The nanograting patterns on the photoactive layer caused an increase in the EQE in the entire wavelength region (especially in the long-wavelength region) owing to the surface plasmon resonance (SPR) effect between the patterned photoactive layer and the metal electrode [19]. Similarly, the SPR effect was exhibited in the ultra-flexible device without nanograting patterns owing to the morphological characteristics of the Ag NWs [19,20].

A compression test was performed to evaluate the mechanical properties of ultra-flexible OPVs. After delamination from the supporting substrate, ultra-flexible OPVs were laminated onto a 200% pre-stretched elastomer film. Then, the devices were compressed in the uniaxial center direction by slowly reducing the stretching force, inducing a tensile strain in the elastomer film of 200–0%, which corresponded to the compression of the OPVs. The PCE decreased from 10.02% to 7.53% after 1000 repetitive compression cycles at 33% compression strain (Figure 7). This excellent durability suggested that ultra-flexible OPVs can be utilized as power sources for mobile devices.

## 4. Conclusions

We have successfully developed an ultra-flexible substrate that can be used for OPVs through a non-vacuum process. Because of the characteristics of CYTOP, which has a lower refractive index than bare glass, it was possible to reduce the optical interference between the substrate and the air. Therefore, the PCE of the OPVs with the ultrathin polymer substrate improved by 6.4% compared to that of the ITO/Glass base OPVs. By introducing the nanograting patterns, it was confirmed that the photocurrent increased cumulatively to achieve a PCE of 10.12%. Moreover, the ultra-flexible substrate showed very good mechanical properties. Therefore, the ultra-flexible substrate fabricated by an easy and inexpensive process can be applied not only to optical devices such as OPVs and organic light-emitting diodes, but also to various fields such as biosensors, transparent heaters, and haptic sensors attached to the human body.

## Figures and Tables

**Figure 1 nanomaterials-10-02185-f001:**
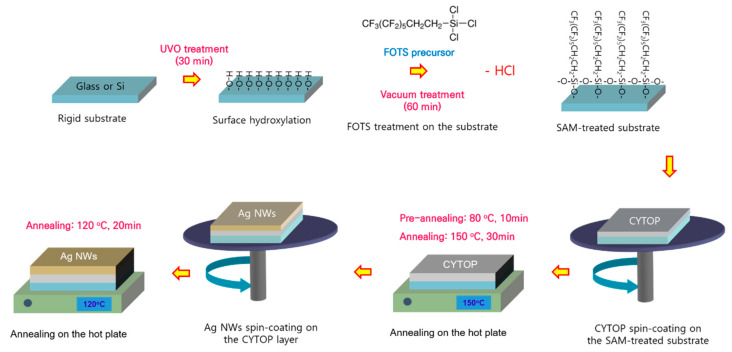
Schematic illustration of the fabrication of the all-solution-processed ultra-flexible substrate with Ag-nanowires transparent electrode.

**Figure 2 nanomaterials-10-02185-f002:**
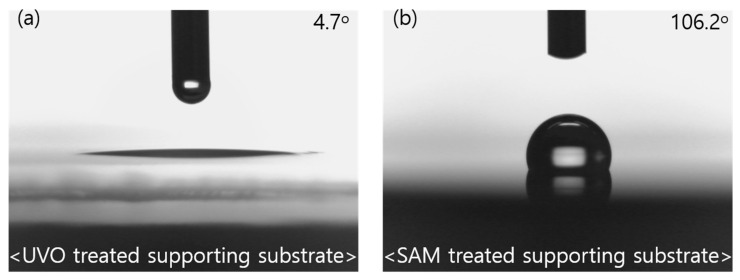
Water contact angle images on supporting substrates of (**a**) UVO treatment; (**b**) trichloro(perfluorooctyl)silane (FOTS) treatment.

**Figure 3 nanomaterials-10-02185-f003:**
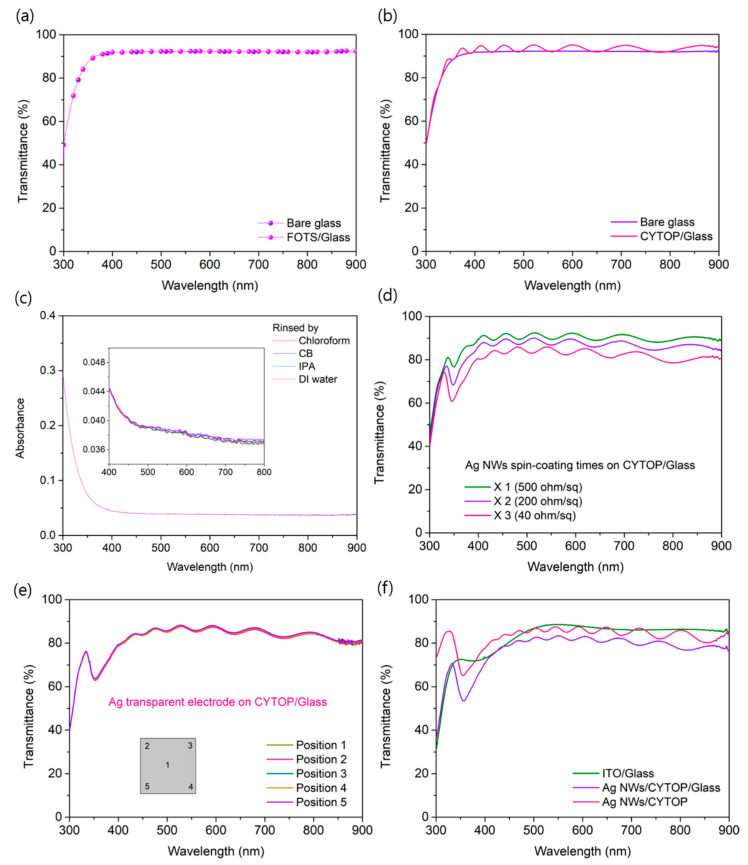
Transmittance characteristics of (**a**) bare glass substrate with and without FOTS treatment; (**b**) CYTOP ultra-flexible substrate on the bare glass substrate; (**c**) solvent resistant test of CYTOP film rinsed with chloroform, chlorobenzene, IPA, and DI water; (**d**) transmittance based on the number of coatings of Ag NWs solution; (**e**) film uniformity of ultra-flexible substrate with Ag NWs/CYTOP/Glass structure; (**f**) various structures with ITO/Glass, Ag NWs/CYTOP/Glass, and Ag NWs/CYTOP freestanding film.

**Figure 4 nanomaterials-10-02185-f004:**
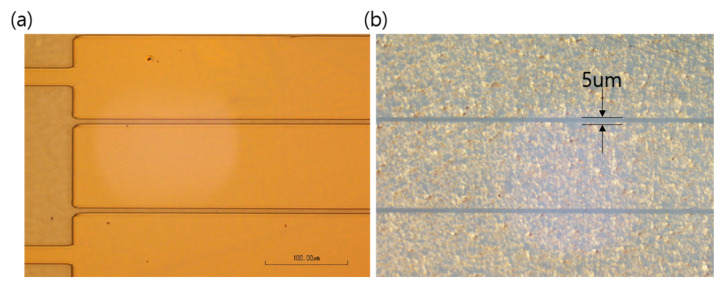
Optical microscopy images: (**a**) After photoresist development process; (**b**) of patterned Ag-nanowires transparent electrode.

**Figure 5 nanomaterials-10-02185-f005:**
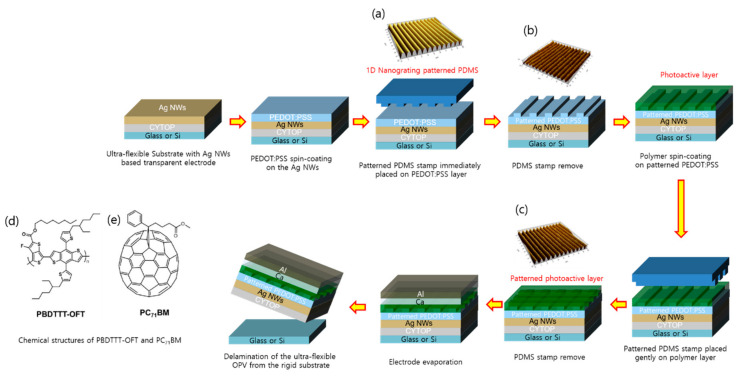
Schematic illustration of the fabrication of ultra-flexible OPVs with a 1D nanograting pattern by soft imprinting lithography using a PDMS mold. Inset figures present: (**a**) 1D nanograting patterned PDMS mold (pitch: 760 nm, depth: 100 nm); (**b**) 1D nanograting patterned PEDOT: PSS layer (pitch: 760 nm, depth: 35 nm); (**c**) 1D nanograting patterned photoactive layer; mixture of PBDTTT-OFT and PC_71_BM (pitch: 760 nm, depth: 55 nm); Chemical structures of (**d**), PBDTTT-OFT and (**e**), PC_71_BM.

**Figure 6 nanomaterials-10-02185-f006:**
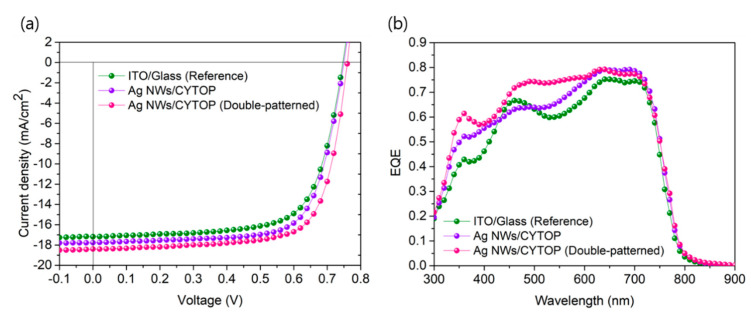
Device characteristics of reference and ultra-flexible organic photovoltaics (OPVs) with various structures: (**a**) *J*-*V* curves under AM 1.5G illumination at 100 mW cm^−2^; (**b**) external quantum efficiency (EQE) spectra of OPVs.

**Figure 7 nanomaterials-10-02185-f007:**
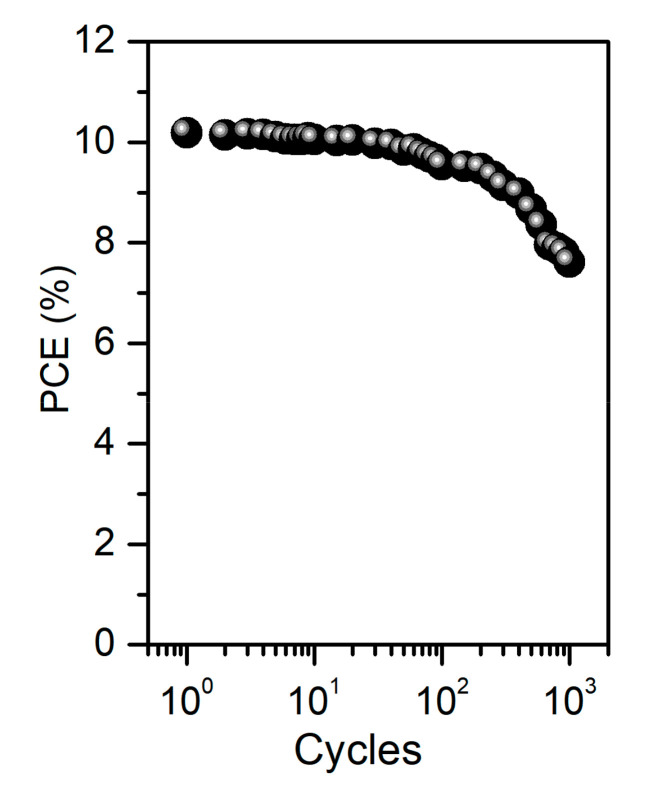
Behavior of photovoltaic power conversion efficiency (PCE) under cyclic compression.

**Table 1 nanomaterials-10-02185-t001:** Device characteristics of various device structures. Values shown in the table are averages over the number of devices indicated in the second column, as follows. R_SH_ and R_S_ are obtained from the slope of the J-V curves under the dark condition at 0 and 1 V, respectively.

Device Structure		*J*_SC_ [mA cm^−2^]	*V*_OC_ [V]	FF [%]	PCE [%]	Increment in *J*_SC_	*R*_Sh_ [kΩ cm^2^]	*R*_S_ [Ω cm^2^]
**PBDTTT-OFT: PC_71_BM (1:1.5)**	Reference ^a^ (ITO/Glass)	17.18 ± 0.01	0.74	69.6 ± 0.2	8.93 ± 0.03	-	5.2	3.07
	Ultra-flexible ^b^	17.77 ± 0.04	0.75	71.3 ± 0.4	9.50 ± 0.1	3.4%	11.8	2.55
	Ultra-flexible ^c^ (Double-patterned)	18.40 ± 0.07	0.76	72.3 ± 0.4	10.12 ± 0.2	7.1	16.6	1.92

^a^ 5 devices; ^b^ 10 devices; ^c^ 12 devices.

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
