# Peer review of "Ultra-Flexible Organic Photovoltaics with Nanograting Patterns Based on CYTOP/Ag Nanowires Substrate"

_nanomaterials, 2020, doi:10.3390/nano10112185_

Round 1
Reviewer 1 Report
Please see the attached file

Author Response
Response to Reviewer 1 Comments
Point-by-point responses to reviewers’ comments
Q1) The manuscript is, in general well written despite some minor typing errors that should be corrected (see below) but I would suggest to avoid the use of “I” in the manuscript, even for a single author
A1) Thanks for your suggestion. Based on your advice, I modified all subjects to "We".
Q2) First of all, I did not find the active area of the OPV cells, neither in the experimental section, nor in the results presentation. This precision is very important in the field as very high PCEs can be measured on very small (and hard to measure) surfaces.
A2) I agree with your opinion. However, I gave the area of ​​the photoactive layer (expressed as the area of ​​the metal mask) in Section 2.5. The active area was 0.12 cm2, as follows.
The active area of each device was defined with a 0.12 cm2 metal photomask.
Q3) Then, in Table 1 were OPV results are presented, there is no indication of the number of diodes used to estimate the photovoltaic parameters and their uncertainties. Without these indications, the presented uncertainties are meaningless.
A3) I agree with your opinion. Table 1 of the submitted manuscript presented the number of measured devices, but the contents were not accurate. I more accurately represented the number of measured devices according to the device structure, as follows.
Table 1. Device characteristics of various device structures. Values shown in the table are averages over the number of devices indicated in the second column, as follows.RSH and RS are obtained from the slope of the J-V curves under the dark condition at 0 and 1 V, respectively
|
Device structure |
JSC [mA cm-2] |
VOC [V] |
FF [%] |
PCE [%] |
Increment in JSC |
RSh [kΩ cm2] |
RS [Ω cm2] |
|
|
PBDTTT-OFT : PC71BM (1:1.5) |
Referencea (ITO/Glass) |
17.18± 0.01 |
0.74 |
69.6 ± 0.2 |
8.93 ± 0.03 |
- |
5.2 |
3.07 |
|
Ultra-flexibleb |
17.77 ± 0.04 |
0.75 |
71.3 ± 0.4 |
9.50 ± 0.1 |
3.4% |
11.8 |
2.55 |
|
|
Ultra-flexiblec (Double-patterned) |
18.40 ± 0.07 |
0.76 |
72.3 ± 0.4 |
10.12 ± 0.2 |
7.1 |
16.6 |
1.92 |
|
a 5 devices
b 10 devices
c 12 devices
Q4) Then, I found the title very ambitious as the final electron-collecting electrode (Ca/Al) is deposited by thermal evaporation under vacuum (Lines 100-102), a process that can hardly be described as “solution-processed”.
A4) I agree with your opinion. However, my meaning of the title was to apply the substrate and transparent electrode as a solution process in a flexible solar cell. Since this meaning may confuse the readers, the title has been revised as follows.
Ultra-flexible organic photovoltaics with nanograting patterns based on CYTOP/Ag nanowires substrate
Q5) Then, I found that the chemical structure of the electron-donor polymer (PBDTTT-OFT) and of the electronacceptor small molecule (PC71BM) was missing.
A5) I agree with your opinion. Figure 4 and caption have been modified as follows.
Figure 4. Schematic illustration of the fabrication of ultra-flexible OPVs with a 1D nanograting pattern by soft imprinting lithography using a PDMS mold. Inset figures present: (a) 1D nanograting patterned PDMS mold (pitch: 760 nm, depth: 100 nm); (b) 1D nanograting patterned PEDOT: PSS layer (pitch: 760 nm, depth: 35 nm); (c) 1D nanograting patterned photoactive layer; mixture of PBDTTT-OFT and PC71BM (pitch: 760 nm, depth: 55 nm); Chemical structures of (d), PBDTTT-OFT and (e), PC71BM.
Q6) Finally, the single author should emphasize the novelty of the present manuscript compared to a previously published work like “Sungjun Park, Soo Won Heo, Wonryung Lee, Daishi Inoue, Zhi Jiang, Kilho Yu, Hiroaki Jinno, Daisuke Hashizume, Masaki Sekino, Tomoyuki Yokota, Kenjiro Fukuda, Keisuke Tajima, Takao Someya, Self-powered ultra-flexible electronics via nanograting-patterned organic photovoltaics, Nature, 561 (2018) p. 516” that deals with the same active-layer, the same grating process, very similar electrodes and process but involved 13 co-authors including Soo Won Heo.
A6) I am very sympathetic to your concerns. However, as mentioned in the introduction, this study has many differences from previous studies. When fabricating an ultra-flexible device, the selection of the ultrathin substrate material and the application of the transparent electrode must be very careful. To the best of my knowledge, I have not seen any ultrathin substrate structure with CYTOP/Ag nanowires in ultra-flexible OPVs. From this point of view, I think the topic of developing the fabrication and process of OPV based on CYTOP/Ag Nanowire is fascinating.
Moreover, the main reason for the introduction of the nanostructure is not only to improve Jsc but also to report that the same process can be applied to the PEDOT: PSS layer.
Based on your advice, the contents of the introduction have been modified as follows.
Organic photovoltaics (OPVs) with a thickness of less than 5 µm are being actively studied as next-generation power sources for mobile applications and biosensors because they are very thin, light, and exhibit very high power-to-weight ratios [1–6]. An important factor in fabricating ultra-flexible OPVs is the development of ultra-flexible polymer substrates and the application of suitable transparent electrode materials.
Many researchers have applied parylene, polydimethylsiloxane (PDMS), and transparent polyimide (PI) as ultrathin polymer substrate materials in ultra-flexible OPVs [1–6]. In the case of parylene, chemical vapor deposition is essential, which makes it difficult to manufacture a large-area device and establish a continuous process [7-9]. For PDMS, solution processes such as spin coating are possible, but the solvent resistance such as acid etchant and alkaline developer is weak, making it difficult to withstand the etching process of transparent conductive oxides such as indium tin oxide (ITO). [10,11]. Furthermore, the transmittance of transparent PI is lower than those of parylene and PDMS, and its chemical resistance and coefficient of thermal expansion values are low [12-14]. Therefore, a more in-depth research is required to develop an ultra-flexible transparent substrate for OPV that satisfies the conditions as mentioned above.
ITO is one of the most widely used transparent electrodes for ultra-flexible OPVs owing to its low sheet resistance and transmittance of more than 90% [15]. However, ITO exhibits very rigid properties owing to its high crystallinity. This characteristic deteriorates the mechanical properties of ultra-flexible OPVs. Therefore, it is very important to introduce material for a transparent electrode with excellent mechanical properties, high transparency, and ease of fabrication.
To overcome this problem, we introduced an amorphous perfluoro-polymer (cyclic transparent optical polymer, CYTOP, AGC chemicals) as a substrate material and fabricated an ultrathin polymer substrate via a solution process [16]. CYTOP has an extremely high transparency, of which the visible light transmission ratio is more than 95% or more, with an amorphous structure completely different from existing fluoropolymers. In addition, it has excellent chemical resistance and moisture-proof property, so it can be applied to ultrathin substrates for ultra-flexible OPVs.
A self-assembly monolayer (SAM) was formed using trichloro(perfluorooctyl)silane (FOTS) to easily delaminate ultra-flexible OPVs on a supporting substrate (bare glass). CYTOP introduced an ultrathin polymer substrate that showed strong solvent resistance to various organic and polar solvents. Thus, we could apply the etching process of the transparent electrode to the ultrathin substrate.
In addition, Ag nanowires (NWs) were introduced as transparent electrodes to improve the mechanical properties of the ultra-flexible OPVs [17,18]. As the number of spin-coatings of the Ag NWs solution increased, the transmittance and sheet resistance of the transparent electrode reached 88% and 40 ohm/sq, respectively. Moreover, 1D nanograting patterns were introduced on both the hole transport layer such as poly(3,4-ethylenedioxythiophene) polystyrene sulfonate (PEDOT: PSS) and the photoactive layer for the first time to increase the power conversion efficiency (PCE) of the ultra-flexible OPV. The PCE improved from 8.93%(rigid structure without pattern) to 10.12% due to an increase in the short circuit current density (JSC) by introducing the 1D nanograting patterns.
Q7) As minor corrections, in Line 67, “Ag NWs (10 mg/ml, Novarials NovaWire-Ag-A70-IPA) was were spin-coated” was should be removed.
A7) Thank you for your careful review. This sentence has been modified as follows.
Ag NWs (10 mg/ml, Novarials NovaWire-Ag-A70-IPA) were spin-coated onto the CYTOP ultrathin polymer layer.
Q8) In Line 98, “suggested that the number of materials would be similar”, “number of” is meaningless and should be replaced by “amount of”.
A8) Thank you for your careful review. This sentence has been modified as follows.
There were no visible remaining photoactive layers on the surface of the PDMS mold after peeling off, which suggested that the amount of materials would be similar in the photoactive layers with or without the patterning.
Q9) Line 140, “I were able”, as already mentioned, I would recommend to drop “I” and certainly to change the sentence in “I was able”.
A9) Thank you for your careful review. This sentence has been modified as follows.
Therefore, as shown in Figure 5, we were able to pattern the Ag NW-based transparent electrode introduced on the CYTOP and secure the lithography process with a resolution of 5 µm.
Q10) Line 147, “was 40 ohm/sq was suitable for the application of OPV” should be changed in “was 40 ohm/sq, a suitable value for the application in OPV”.
A10) Thank you for your careful review. This sentence has been modified as follows.
The sheet resistance of the Ag NWs that was coated three times was 40 ohm/sq, a suitable value for the application in OPVs.
Q11) In Figure 4, line (c), the three first drawings indicate “polymer” or “patterned polymer” while it concerns the active layer: a electron-donor polymer and an electron-acceptor small molecule
A11) Thank you for your careful review. Figure 4 and caption have been modified as follows.
Figure 4. Schematic illustration of the fabrication of ultra-flexible OPVs with a 1D nanograting pattern by soft imprinting lithography using a PDMS mold. Inset figures present: (a) 1D nanograting patterned PDMS mold (pitch: 760 nm, depth: 100 nm); (b) 1D nanograting patterned PEDOT: PSS layer (pitch: 760 nm, depth: 35 nm); (c) 1D nanograting patterned photoactive layer; mixture of PBDTTT-OFT and PC71BM (pitch: 760 nm, depth: 55 nm); Chemical structures of (d), PBDTTT-OFT and (e), PC71BM.
Reviewer 2 Report
The findings of this manuscript will be interesting for OPV scientists working on flexible solar cells. The PCE of >10% achieved on flexible substrates is quite notable as well. A drop in 33% of efficiency after 1000 stretching cycles is not great, but it is acceptable since it is an extreme test. Hence, I recommend publication of this manuscript with a few minor points to be addressed:
Line 189 states that Rsh is reduced. From the sentence, it sounds like the nano patterned cells reduced Rsh, contrary to what is shown in table 1, an increase in Rsh as compared to ITO/glass reference. Please clarify.
Lines 203-205: morphology of Ag NWs creates SPR effect that leads to enhancement in EQE and hence contributes to increase in Jsc. Is this something well known? If yes, please add citation.
Author Response
Response to Reviewer 2 Comments
The findings of this manuscript will be interesting for OPV scientists working on flexible solar cells. The PCE of >10% achieved on flexible substrates is quite notable as well. A drop in 33% of efficiency after 1000 stretching cycles is not great, but it is acceptable since it is an extreme test. Hence, I recommend publication of this manuscript with a few minor points to be addressed:
Q1) Line 189 states that Rsh is reduced. From the sentence, it sounds like the nano patterned cells reduced Rsh, contrary to what is shown in table 1, an increase in Rsh as compared to ITO/glass reference. Please clarify.
A1) Thank you for your careful review. This sentence is wrong by my mistake. This sentence has been modified as follows.
The introduction of nanograting patterns slightly decreased the RS values compared with the reference device, whereas it was increased the RSh values [19].
Q2) Lines 203-205: morphology of Ag NWs creates SPR effect that leads to enhancement in EQE and hence contributes to increase in Jsc. Is this something well known? If yes, please add citation.
A2) Thank you for your question. We conducted a study on the origin of increasing Jsc in a previous study. In the last study, the SPR effect occurred on the patterned polymer surface and metal electrode. However, in this study, the same result could be obtained even in a device without a pattern. (The increase in EQE in the region after 730nm in Figure 6(b) proves this.) As the reviewer asked, it is a well-known fact that the SPR effect of organic solar cells incorporating Ag NWs has occurred, and the following reference support this.
- Huang, P.-C.;Chen, T.-Y.; Wang, Y.-L.; Wu, C.-Y.; Lin, T.-L. Improving interfacial electron transfer and light harvesting in dye-sensitized solar cells by using Ag nanowire/TiO2 nanoparticle composite films. RSC Adv. 2015, 5, 70172-70177
And, the previous sentence has been modified as follows.
Similarly, the SPR effect was exhibited in the ultra-flexible device without nanograting patterns owing to the morphological characteristics of the Ag NWs [19,20].

Reviewer 3 Report
- Summary of the research and overall impression:
Focus, Aim and Key data:
The manuscript focuses on the development of a method for fabricating ultrathin polymer substrates for ultra-flexible organic photovoltaics (OPVs). The aim of the study was to replace commonly used substrate materials, examine the overall behavior of the system as an OPV and evaluate the mechanical properties. The author’s arguments are supported by experimental data, with key data the mechanical properties of the system.
Summary of the manuscript:
The author starts from the motivation of the study, which stems from the ineffective utilization of substrate materials for ultra- flexible OPVs applications. Subsequently, the author explained the methods performed and the materials utilized in his study. In the discussion part, the author supported his scientific claims with experimental data. The approach of the author was to examine, the optical characteristics of the materials, then the electrical characteristics of the OPV systems and then determine the mechanical properties. The author concluded that the development of his claim was successful and emphasized the adequacy of the developed substrate for ultra-flexible electronics mobile applications.
Overall impression:
The author’s approach is sensible and concise with scientific accuracy. The timeliness is sufficient with more than 60% of the author’s references being published after 2016. The authoritativeness is plentiful covering citations focused on high transparent and ultra- flexible materials for relevant applications. The main strengths of this manuscript is that it addresses an interesting topic, finds a novel solution based on a carefully selected set of rules, and provides an answer. On the other hand, a lot of emphasis is given on the 1-D nanograting, which is interesting, but may be considered beyond the scope of the manuscript and confuse the reader. Furthermore, the author may wish to include or expand more on the mechanical properties of the developed substrate and its potential exploitation either by industry or research. No other significant faults were detected, but a number of minor points, which the author may wish to revise before publication.
Recommended decision:
Accept with major revision.
Comment: The scientific accuracy and sensible approach of the study are well established, as well as interesting with a novel solution. However, the author may wish to consider a major revision due to the number of minor issues detected. Furthermore, the author may wish to emphasize further on the novelty of the study and the impact of his results in related applications.
- Discussion of specific areas for improvement:
Title and cover:
- The title is short, on point and gives a clear idea of the manuscript’s content
Abstract and Keywords:
- The abstract is well structured and summarizes the manuscript
- Line 12 “and a resolution…process”: The author may wish to exclude it from the abstract, as it may appear not relevant with the abstract.
- The author may wish to state even clearer the focus and aim of the manuscript. It is encouraged to utilize words like aim of the study, scope of the paper.
- Line 18 “PCE.”: The author may wish to avoid the double-use of the abbreviation “PCE”. Suggested wording: “initial value”
Introduction:
- The author correctly states the problem coming from the utilization of conventional materials and proposes a solution
- Lines 49-53 “As the… to 10,12%”: The author may wish to include this in the discussion section and not in the introduction.
- The citation are referenced correctly
Materials and Methods:
- The author provides a good explanation of the materials used and methodology followed
- Line 71- 102: The author may wish to consider shortening this paragraph, since this procedure is not the main focus of the innovative solution. As the title suggests the novelty is focused on the “all solution processed substrate”.
Results- Discussion:
- Well-structured approach and discussion, with sensible scientific argumentation. The author correctly points out the strong points of his study. In addition, it is pleasurable to see that many additional angles were considered like the solvent resistance and film uniformity.
- Line 141: The author may wish to include how he realized the 5μm resolution structure.
- Line 189: The author states that the Rsh was reduced, but in table the value was increased compared to the reference device. The author may wish to revise this.
- Lines 119- 205: The author’s arguments are point and sensible, but this paragraph averts the reader’s attention to the nanograting patterns, rather the substrate. It is suggested to focalize more on the novelty of the substrate for the specific application.
Conclusions:
- No new data are included in the conclusions, which is correct. The data considered are mentioned and reflected upon, clearly in the discussion part and the main findings are summarized sufficiently.
- The author may wish to revise the conclusion section, as it almost the same as in the abstract. It is encouraged to rephrase a few sentences to avoid repetition.
- The author may wish to include the effect of the substrate on the transmittance as well, for a more complete sum-up and to emphasize more on the utilization of the said substrate. This also assists the reader to realize more effectively why the solution proposed is suitable for ultra-flexible OPVs.
- The author may wish to include additional or more specific potential applications/innovations, to emphasize more on the importance of this novelty. This helps the reader to clear the “take-home” message
References:
- The references are cited correctly and are relevant to the author’s study
- No issues/faults detected during cross-validation of the citations
Tables:
- Table 1: The RSh value of the Ultra-flexible (Double-patterned) was increased compared to the reference device. In the text (Line: 189), it is stated by the author that it was “reduced”. The author may wish to revise.
Figures:
- Figure 1: A few of the texts below the images appear intercepted by the image and in poor quality. Suggestion to author: Reposition the text and re-align. A good trick might also be to use a little larger size letter.
- Figure 3:
- Figure 3(b): The author may wish to include the “Bare glass” in the captcha of the figure
- Figure 3(c): The author may wish to remove one of the legends of the graph since they appear twice.
- Figure 4:
- Figure 4(c) (Image in the middle): The author may wish to reposition or remove the word “Polymer”, in red text, since it is not visible there and the reader is already informed from the image before that.
Other:
- Author contributions, Funding and Conflict of interested are stated sufficiently. This shows the hard work of the author and pays tribute to the foundation and institute that supported his study.
Author Response
Response to Reviewer 3 Comments
Recommended decision:
Accept with major revision.
Comment: The scientific accuracy and sensible approach of the study are well established, as well as interesting with a novel solution. However, the author may wish to consider a major revision due to the number of minor issues detected. Furthermore, the author may wish to emphasize further on the novelty of the study and the impact of his results in related applications.
Title and cover:
Q1) The title is short, on point and gives a clear idea of the manuscript’s content
A1) Thank you for your thoughtful review. However, one of the reviewers recommended that the title be changed to reflect my manuscript's overall content. To reduce the possibilities of confusion for readers, the title has been changed as follows.
Ultra-flexible organic photovoltaics with nanograting patterns based on CYTOP/Ag nanowires substrate
Abstract and Keywords:
The abstract is well structured and summarizes the manuscript;
Q2) Line 12 “and a resolution…process”: The author may wish to exclude it from the abstract, as it may appear not relevant with the abstract.
A2) Thank you for your suggestion. Based on your advice, I revised the above sentence from the abstract as follows.
In addition, a Ag nanowire was used as a transparent electrode in a solution process.
The author may wish to state even clearer the focus and aim of the manuscript. It is encouraged to utilize words like aim of the study, scope of the paper;
Q3) Line 18 “PCE.”: The author may wish to avoid the double-use of the abbreviation “PCE”. Suggested wording: “initial value”
A3) Thank you for your suggestion. Based on your advice, I revised the above sentence in the abstract as follows.
I performed 1000 cycles of compression/relaxation tests to evaluate the mechanical properties of the ultra-flexible OPV, after which, the PCE remained at 67% of the initial value.
Introduction:
The author correctly states the problem coming from the utilization of conventional materials and proposes a solution;
Q4) Lines 49-53 “As the… to 10.12%”: The author may wish to include this in the discussion section and not in the introduction.
A4) Thank you for your suggestion. Based on your advice, I revised the above paragraph in the introduction and modified the discussion part as follows.
“As the number of spin-coatings of the Ag NWs solution increased, the transmittance and sheet resistance of the transparent electrode reached 88% and 40 ohm/sq, respectively.”->Deleted and modified as follows.
Lines 149-151 “The sheet resistance of the Ag NWs that was coated three times was 40 ohm/sq. Moreover, the transmittance of the Ag NW-based transparent electrode reached 88%, it was suitable for the application of OPV as a transparent electrode.”
“Moreover, the PCE of double-patterned ultra-flexible OPVs with a 1D nanograting pattern on both the electron transport layer and the photoactive layer increased from 8.93% (rigid structure) to 10.12%.” ->Deleted
The citation are referenced correctly
Materials and Methods:
The author provides a good explanation of the materials used and methodology followed
Q5) Line 71- 102: The author may wish to consider shortening this paragraph, since this procedure is not the main focus of the innovative solution. As the title suggests the novelty is focused on the “all solution processed substrate”.
A5) Thanks for your suggestion, and I agree with your opinion. However, because the manuscript title has been changed, I think it is necessary to address the experimental method for this part.
Results- Discussion:
Well-structured approach and discussion, with sensible scientific argumentation. The author correctly points out the strong points of his study. In addition, it is pleasurable to see that many additional angles were considered like the solvent resistance and film uniformity.
Q6) Line 141: The author may wish to include how he realized the 5μm resolution structure.
A6) Thank you for your advice. As your comment, I have revised the section 2.3 as follows.
To form a pattern of Ag NWs based transparent electrode, photoresist (AZ 1512, Microchem) was spin-coated in two steps (500 rpm for 20 s, 3000 rpm for 60 s), and soft bake was performed at 90°C for 1 min on a hot plate. And then, prepared sample was exposed to UV light for 8 s using mask aligner (UFM-50110R, Yamashita Denso). The exposed photoresist was removed by an alkaline tetra methyl ammonium hydroxide developer such as AZ300 (Microchem). The sample was baked at 110°C for 1 min after development to further cross-link the phenolic resin in the unexposed resist. The developed sample was etched by etchant for Ag NWs (Pure Etch GNW300, Hayashi Pure Chemical). The etched sample was rinsed with deionized water, and we could achieved the patterned structure with 5 μm resolution.
Q7) Line 189: The author states that the Rsh was reduced, but in table the value was increased compared to the reference device. The author may wish to revise this.
A7) Thank you for your careful review. This sentence is wrong by my mistake. This sentence has been modified as follows.
The introduction of nanograting patterns slightly decreased the RS values compared with the reference device, whereas it was increased the RSh values [19].
Q8) Lines 119- 205: The author’s arguments are point and sensible, but this paragraph averts the reader’s attention to the nanograting patterns, rather the substrate. It is suggested to focalize more on the novelty of the substrate for the specific application.
A8) Thank you for your careful review, and I agree with your opinion. I think the answer to this question is related to my answer to question 1. As you already know, what I am talking about in this paper is focused on ultra-flexible substrates and transparent electrodes that are manufactured through a non-vacuum process. Moreover, it was also to show that the introduction of the previously applied nanostructure on the newly developed substrate thus produced shows significant results. Therefore, I have inserted the following sentences to explain the reason.
We fabricated OPVs incorporating nanostructures to confirm that substrates fabricated with a non-vacuum process can increase the photocurrent cumulatively without optical interference. In this case, since the Ag NWs was used as the transparent electrode, PEDOT: PSS was introduced as the hole transport layer, unlike the device structure applied in our previous study [19].
Conclusions:
Q9) No new data are included in the conclusions, which is correct. The data considered are mentioned and reflected upon, clearly in the discussion part and the main findings are summarized sufficiently. The author may wish to revise the conclusion section, as it almost the same as in the abstract. It is encouraged to rephrase a few sentences to avoid repetition. The author may wish to include the effect of the substrate on the transmittance as well, for a more complete sum-up and to emphasize more on the utilization of the said substrate. This also assists the reader to realize more effectively why the solution proposed is suitable for ultra-flexible OPVs. The author may wish to include additional or more specific potential applications/innovations, to emphasize more on the importance of this novelty. This helps the reader to clear the “take-home” message
A9) Thank you for your careful review, and I agree with your opinion. Based on your opinion, I have modified as follows.
We have successfully developed an ultra-flexible substrate that can be used for OPVs through a non-vacuum process. Because of the characteristics of CYTOP, which has a lower refractive index than bare glass, it was possible to reduce the optical interference between the substrate and the air. Therefore, the PCE of the OPVs with the ultrathin polymer substrate improved by 6.4% compared to that of the ITO/Glass base OPVs. By introducing the nanograting patterns, it was confirmed that the photocurrent increased cumulatively to achieve a PCE of 10.12%. Moreover, the ultra-flexible substrate showed very good mechanical properties. Therefore, the ultra-flexible substrate fabricated by an easy and inexpensive process can be applied not only to optical devices such as OPVs and organic light-emitting diodes, but also to various fields such as biosensors, transparent heaters, and haptic sensors attached to the human body.
References:
The references are cited correctly and are relevant to the author’s study
No issues/faults detected during cross-validation of the citations
Tables:
Q10) Table 1: The RSh value of the Ultra-flexible (Double-patterned) was increased compared to the reference device. In the text (Line: 189), it is stated by the author that it was “reduced”. The author may wish to revise.
Q10) Thank you for your careful review. This sentence is wrong by my mistake. This sentence has been modified as follows.
The introduction of nanograting patterns slightly decreased the RS values compared with the reference device, whereas it was increased the RSh values [19].
Figures:
Q11) Figure 1: A few of the texts below the images appear intercepted by the image and in poor quality. Suggestion to author: Reposition the text and re-align. A good trick might also be to use a little larger size letter.
A11) Thank you for your careful review. This figure 3(c) has been modified as follows.
Q12) Figure 3(b): The author may wish to include the “Bare glass” in the captcha of the figure
Q12) Thank you for your careful review. This caption of Figure3(b) has been modified as follows.
(b) CYTOP ultra-flexible substrate on the bare glass substrate;
Q13) Figure 3(c): The author may wish to remove one of the legends of the graph since they appear twice.
A13) Thank you for your careful review. This legends of the figure 3(c) has been modified as follows.
Q14) Figure 4(c) (Image in the middle): The author may wish to reposition or remove the word “Polymer”, in red text, since it is not visible there and the reader is already informed from the image before that.
A14) Thank you for your careful review. This figure 3(c) has been modified as follows. In addition, one of the reviewers recommended inserting the chemical structure of the polymer and PC71BM.
Other:
Author contributions, Funding and Conflict of interested are stated sufficiently. This shows the hard work of the author and pays tribute to the foundation and institute that supported his study.
Round 2
Reviewer 1 Report
The manuscript has been substantially improved but I’m still very much concerned about the possibility to conduct such a study by a single author.
This beeing said, I'll focus on errors that should be corrected.
L11: “In addition, a Ag nanowire was used as a transparent electrode” is not understandable. A Ag nanowire is a nano-object that can not be used as an electrode. The sentence should be replaced by “In addition, a Ag nanowire network was used as a transparent electrode” or some equivalent sentence.
L12: “All processes were conducted via spin coating in large area” seems to me incorrect. I would rather use ““All processes were conducted on large area via spin coating”
L72: “we could achieved” is incorrect and should be replaced by “we could achieve”.
L104: “that the number of materials” is meaningless. “The amount of materials” refers to a quantity, not a number.
L147: “Figure 5” is used for the first tile before Figure 4. Therefore, the figure order (and consequently number) should be changed.
L147: “I were able” should be changed by “I was able”.
Figure 4: in this figure, the photoactive layer is referred as “Polymer” (last figure in the first row) and “Patterned polymer” (below (c) in the second row). This denomination is not correct as in both cases, its “photoactive layer” or “patterned photoactive layer”.
L184: Figure 5 is in fact Figure 4.
L203: The RSh values are increased, not reduced by the introduction of nanograting patterns.
Figure 7: a straight line appears below the figure.
Author Response
Response to Reviewer 1 Comments
Comments and Suggestions for Authors:
The manuscript has been substantially improved but I’m still very much concerned about the possibility to conduct such a study by a single author.
This beeing said, I'll focus on errors that should be corrected.
Q1) L11: “In addition, a Ag nanowire was used as a transparent electrode” is not understandable. A Ag nanowire is a nano-object that can not be used as an electrode. The sentence should be replaced by “In addition, a Ag nanowire network was used as a transparent electrode” or some equivalent sentence.
A1) Thank you for your suggestion. Based on your advice, I modified this sentence as follows.
In addition, a Ag nanowire network layer was used as a transparent electrode in a solution process.
Q2) L12: “All processes were conducted via spin coating in large area” seems to me incorrect. I would rather use ““All processes were conducted on large area via spin coating”
A2) Thank you for your suggestion. Based on your advice, I modified this sentence as follows.
All processes were conducted on large area via spin coating.
Q3) L72: “we could achieved” is incorrect and should be replaced by “we could achieve”.
A3) Thank you very much for your meticulous review. I modified this sentence as follows.
The etched sample was rinsed with deionized water, and we could achieve the patterned structure with 5 μm resolution.
Q4) L104: “that the number of materials” is meaningless. “The amount of materials” refers to a quantity, not a number.
A4) I also agree with your opinion. Therefore, the sentence of the latter part have been deleted and modified as follows.
There were no visible remaining photoactive materials on the surface of the PDMS mold after peeling off.
Q5) L147: “Figure 5” is used for the first tile before Figure 4. Therefore, the figure order (and consequently number) should be changed.
Q6) L184: Figure 5 is in fact Figure 4.
A5 and 6) Thank you very much for your meticulous review. I changed the order and position of Figures 4 and 5. Also, I modified some parts of main text and figure captions as follows.
Main text:
To prepare the photoactive layer, we dissolved a mixture of poly[4,8-bis(5-(2-ethylhexyl)thiophen-2-yl)benzo[1,2-b;4,5-25b’]dithiophene-2,6-diyl-alt-(4-octyl-3-fluorothieno [3,4-b]thiophene)-2-carboxylate-2-6-diyl]26 (PBDTTT-OFT) as an electron donor polymer and [6,6]-phenyl-C71-butyric acid methyl ester (PC71BM) as an electron acceptor small molecule (1: 1.5, 10 mg: 15 mg, Figure 5(d) and (e)) in 1 mL chlorobenzene with 3 vol% 1,8-diiodooctane (98%, Merck).
Therefore, as shown in Figure 4, we were able to pattern the Ag NW-based transparent electrode introduced on the CYTOP and secure the lithography process with a resolution of 5 µm using AZ1512 positive photoresist (Microchem).
Figure 5 exhibits a schematic illustration of the fabrication of ultra-flexible OPVs with a 1D nanograting pattern.
Figure caption
Figure 4. Optical microscopy images: (a) after photoresist development process; (b) of patterned Ag-nanowires transparent electrode.
Figure 5. Schematic illustration of the fabrication of ultra-flexible OPVs with a 1D nanograting pattern by soft imprinting lithography using a PDMS mold. Inset figures present: (a) 1D nanograting patterned PDMS mold (pitch: 760 nm, depth: 100 nm); (b) 1D nanograting patterned PEDOT: PSS layer (pitch: 760 nm, depth: 35 nm); (c) 1D nanograting patterned photoactive layer; mixture of PBDTTT-OFT and PC71BM (pitch: 760 nm, depth: 55 nm); Chemical structures of (d), PBDTTT-OFT and (e), PC71BM.
Q7) L147: “I were able” should be changed by “I was able”.
A7) Thank you very much for your meticulous review. All subjects have been changed to "We".
Q8) Figure 4: in this figure, the photoactive layer is referred as “Polymer” (last figure in the first row) and “Patterned polymer” (below (c) in the second row). This denomination is not correct as in both cases, its “photoactive layer” or “patterned photoactive layer”.
A8) Thank you for your advice. I revised this figure as follows.
Q9) L203: The RSh values are increased, not reduced by the introduction of nanograting patterns.
A9) Thank you very much for your meticulous review. I have already revised this part in the previous revised version as follows.
The introduction of nanograting patterns slightly decreased the RS values compared with the reference device, whereas it increased the RSh values [19].
Q10) Figure 7: a straight line appears below the figure.
A10) Thank you for checking the figure. I modified this as follows.
Figure 7. Behavior of photovoltaic PCE under cyclic compression.

Reviewer 3 Report
Comments and Suggestions for Authors:
This is an interesting topic and many thanks to the author and the journal that gave me the opportunity to review his manuscript. The author took into consideration all of my comments and the proposed areas for improvement and revised all relevant points. I agree that the tittle change was a sensible approach and that resolved any issues regarding the reader’s confusion. More specifically, the title change, the revision of the conclusion and other minor parts clarified and covered the main points of my previous review. The article is well-written, treats an actual problem and provides evidence about a novel solution. The scientific accuracy and approach are well established, and it was a very interesting topic in the field of flexible organic photovoltaics.
Author Response
Response to Reviewer 3 Comments
Comments and Suggestions for Authors:
Q1) This is an interesting topic and many thanks to the author and the journal that gave me the opportunity to review his manuscript. The author took into consideration all of my comments and the proposed areas for improvement and revised all relevant points. I agree that the tittle change was a sensible approach and that resolved any issues regarding the reader’s confusion. More specifically, the title change, the revision of the conclusion and other minor parts clarified and covered the main points of my previous review. The article is well-written, treats an actual problem and provides evidence about a novel solution. The scientific accuracy and approach are well established, and it was a very interesting topic in the field of flexible organic photovoltaics.
A1) Thank you for your heartfelt congratulations and encouragement.
